# Mechanistic Insights into the Neuroprotective Potential of Sacred *Ficus* Trees

**DOI:** 10.3390/nu14224731

**Published:** 2022-11-09

**Authors:** Kyu Hwan Shim, Niti Sharma, Seong Soo A. An

**Affiliations:** Department of Bionano Technology, Gachon University, 1342 Seongnam-daero, Sujeong-Gu, Seongnam 461-701, Korea

**Keywords:** *F. religiosa*, *F. benghalensis*, neurodegenerative disorders, bioactive compounds, multi-target approach

## Abstract

*Ficus religiosa* (Bo tree or sacred fig) and *Ficus benghalensis* (Indian banyan) are of immense spiritual and therapeutic importance. Various parts of these trees have been investigated for their antioxidant, antimicrobial, anticonvulsant, antidiabetic, anti-inflammatory, analgesic, hepatoprotective, dermoprotective, and nephroprotective properties. Previous reviews of *Ficus* mostly discussed traditional usages, photochemistry, and pharmacological activities, though comprehensive reviews of the neuroprotective potential of these *Ficus* species extracts and/or their important phytocompounds are lacking. The interesting phytocompounds from these trees include many bengalenosides, carotenoids, flavonoids (leucopelargonidin-3-O-β-d-glucopyranoside, leucopelargonidin-3-O-α-l-rhamnopyranoside, lupeol, cetyl behenate, and α-amyrin acetate), flavonols (kaempferol, quercetin, myricetin), leucocyanidin, phytosterols (bergapten, bergaptol, lanosterol, β-sitosterol, stigmasterol), terpenes (α-thujene, α-pinene, β-pinene, α-terpinene, limonene, β-ocimene, β-bourbonene, β-caryophyllene, α-trans-bergamotene, α-copaene, aromadendrene, α-humulene, alloaromadendrene, germacrene, γ-cadinene, and δ-cadinene), and diverse polyphenols (tannin, wax, saponin, leucoanthocyanin), contributing significantly to their pharmacological effects, ranging from antimicrobial action to neuroprotection. This review presents extensive mechanistic insights into the neuroprotective potential, especially important phytochemicals from *F. religiosa* and *F. benghalensis*. Owing to the complex pathophysiology of neurodegenerative disorders (NDDs), the currently existing drugs merely alleviate the symptoms. Hence, bioactive compounds with potent neuroprotective effects through a multitarget approach would be of great interest in developing pharmacophores for the treatment of NDDs.

## 1. Introduction

Neurodegeneration or neuronal atrophy results from progressive degeneration of neuronal structures and/or functions. It predominantly affects the elderly population as a spectrum of neurodegenerative disorders (NDDs), including Alzheimer’s disease (AD), Parkinson’s disease (PD), Huntington’s disease (HD), and amyotrophic lateral sclerosis (ALS). NDDs lead to a progressive decline in mental (cognition, memory, orientation, attention) and motor movements (gait, orientation, balance). Neurodegeneration affects diverse neuronal circuitries at different locations in the brain. These alterations lead to the loss of dopaminergic neurons (PD), dorsal striatum neurons (HD), both upper and lower motor neurons (ALS), and axonal neuron degeneration (temporal and hippocampal neurons in AD) [1,2,3,4]. Aggregation of proteins and peptides into their pathological forms are the key events in NDDs [5], for example, amyloid beta protein (Aβ) and Tau in AD [6]; mutant huntingtin (mHTT) protein in HD [7]; α-synuclein (αSN) in PD and TAR DNA-binding protein 43 (TDP-43) in ALS [8]. Other factors, such as neuroinflammation, oxidative stress, age, genetics, and environmental factors, also contribute to the etiopathology of NDDs [9,10]. No doubt, tremendous efforts are poured in to find the ways for early diagnoses and cures for NDDs, but only little advances have been achieved in both avenues. Recently, groundbreaking advances have been made regarding early blood diagnoses of several NDDs by targeting genetic mutations, Aβ oligomers, and Aβ42 [11,12,13,14,15]. The existing therapies target only a specific pathway/enzyme and are suitable to alleviate only the symptoms, not stop the disease progression or disease modifications. Since many medicinal plants have been documented in various traditional texts for their therapeutic effects, several compounds for treating NDDs are of plant origin, such as levodopa (in PD treatment) and galantamine (in AD treatment) obtained from *Vicia faba* [16] and *Galanthus* [17], respectively. Hence, a search for bioactive compounds that can exert a neuroprotective effect through a multitarget approach is desirable. In addition, with the possibility of detecting various oligomers of NDDs in blood by a multimer detection system (MDS), screening bioactive compounds from plant extracts would be feasible. Currently, we are investigating neuroprotective mechanism of selected extracts in our laboratory using a combination of chemical, biochemical, molecular, and cell-based assays. The data obtained are quite promising and we will be publishing the results soon.

The genus *Ficus* (family Moraceae) consists of over 850 species of vines, shrubs, and trees, occupying a larger part of the tropical and subtropical forest ecosystem [18,19]. *Ficus* trees are among the highest oxygen producers in nature with a prime photosynthesis rate as well as a rich source of mineral deposits in the leaves [20]. Among the *Ficus* species, *Ficus religiosa* (Bo tree or sacred fig) and *Ficus benghalensis* (Indian banyan) have a long history of spiritual significance in the Indian subcontinent. As the name suggests, *F. religiosa* is considered a spiritual tree in Buddhism and Hinduism, as Gautama Buddha attained enlightenment under this tree in India, and Buddhism originated. In 288 BCE, a cutting from this tree was planted in Sri Lanka and the tree is still alive, which gives an astonishing account of the age of the tree. *F. benghalensis* is the national tree of India, symbolizing “wisdom and eternity” [21]. Several references to *Ficus* are also mentioned in the Bible where Adam and Eve used *Ficus* leaves to cover their bodies in heaven (Genesis 3:7) [22]. All these references signify the importance and eternity of the *Ficus* tree.

*F. religiosa* is a deciduous tree with heart-shaped leaves, while *F. benghalensis* is evergreen, with leathery and ovate leaves. In the Indian traditional system of medicine (Ayurveda), different parts of these trees are used to treat cough, asthma, heart diseases, nose bleeding, diabetes, toothaches, constipation, fever, jaundice, wounds, gonorrhea, and skin infections [23,24,25]. These trees present a wide range of promising bioactive compounds, such as phenols, flavonoids, carotenoids, sterols, anthocyanins, alkaloids, tannins, saponins, terpenoids, and vitamins, possessing a wide range of biological properties. [18]. *F. benghalensis* and *F. religiosa* share several common therapeutic activities, such as antiulcerogenic [26,27], anticancer [28,29], antidiabetic [30], antipyretic [31,32], hypolipidemic [33,34], anthelminthic [35], anti-inflammatory [32,36], and immunomodulatory properties [37,38]. Apart from this, *F. benghalensis* also displays antidiarrheal [39] and antiallergic activities [40], while *F. religiosa* boasts bronchodilatory, anti-asthmatic [41], and anticonvulsant properties [42].

Previous reviews of these species mostly discussed the traditional usages, photochemistry, and pharmacological activities; however, comprehensive reviews of the neuroprotective potential of these *Ficus* species extracts and/or their important phytocompounds are lacking. This report presents an inclusive review of the existing scientific works from various databases (PubMed, Google Scholar, and Science Direct) published before July 2022 on the neuroprotective mechanisms of *F. religiosa* and *F. benghalensis* extracts, as well as the major bioactive compounds present therein.

The keywords in English included “neuroprotection”, “*Ficus religiosa*”, “*Ficus benghalensis*”, “Alzheimer’s disease”, “neurodegenerative diseases”, “extracts”, and “bioactive metabolites”. All the literature about in vitro, in vivo, and in silico studies related to protein aggregation mechanism, oxidative stress, antioxidant parameters, proinflammatory cytokines, enzyme inhibition, metabolic pathways, gene expression, neurotransmitter levels, memory, and cognition were included. Literature in a language other than English and unpublished works were excluded. 

## 2. Phytochemicals and Therapeutic Properties

### 2.1. F. religiosa

All parts of the plant (roots, bark, fruits, leaves, latex, and seeds) are of therapeutic importance [43] due to the presence of several important phytochemicals. The bark of *F. religiosa* contains high concentrations of polyphenols (tannin, wax, saponin, leucoanthocyanin), flavonoids (leucopelargonidin-3-O-β-d-glucopyranoside, leucopelargonidin-3-O-α-l-rhamnopyranoside, lupeol, cetyl behenate, and α-amyrin acetate), flavonols (kaempferol, quercetin, myricetin), and polysterols (bergapten, bergaptol, lanosterol, β-sitosterol, stigmasterol) [44,45]. The major compounds (lupenol, γ-sitosterol and 1,2-benzenediol) identified in the stem bark extract (petroleum ether, chloroform, and methanol) by GC–MS [46] display anti-inflammatory, antidiarrheal, hypoglycemic, and antibacterial properties [30,47].

The leaves are a rich source of polyphenols (eugenol, hexanol, phytol), sesquiterpene (eudesmol), and several other compounds, such as α-copaene, linalool, salicylaldehyde, phenylacetaldehyde, allyl caproate, n-nonanal, adipoin, methylcyclopentane, 2-dione, itaconic anhydride, benzeneacetonitrile, nonadienal, nonadienol, catechol, coumarin, cinnamyl alcohol, vinyl guaiacol, α-cadinol, pentadecanal, and palmitic acid [48]. The leaves are used to relieve nose bleeding and blood in urine and stool due to coagulative/anti-fibrinolytic properties. Additionally, the leaf juice is used for treating asthma, migraine, toothache, diarrhea, wounds, and gastric disorders [49].

The fruits have laxative properties [25] and contain abundant flavonols (quercetin, kaempferol, myricetin), terpenes/terpenoids (α-thujene, α-pinene, β-pinene, α-terpinene, limonene, β-ocimene, β-bourbonene, β-caryophyllene, α-trans-bergamotene, α-copaene, aromadendrene, α-humulene, alloaromadendrene, germacrene, γ-cadinene, and δ-cadinene), and polyphenols (stigmasterol, lupeol) [50,51,52,53]. The *F. religiosa* latex is used to treat neuralgia and contains serine proteases (religiosin B and C) [54]. The seeds contain high concentrations of alanine, tyrosine, and threonine amino acids [55].

### 2.2. F. benghalensis

In traditional medicine, the *F. benghalensis* root and bark are used as antidiabetic, anti-inflammatory, antidiarrheal [56], cholesterol-lowering [57], anthelmintic [35], and anti-asthmatic medications [40]. The leaves seem to boost the immune system and are used to treat leucorrhoea [58]. The seeds and latex are used in treating peptic ulcers [59] and urinary disorders [60], respectively.

The important phytocomponents of the *F. benghalensis* leaves are rutin, β-amyrin, leucopelargonin, bengalenoside, psoralen, β-sitosterol, and bergapten [61], while β-sitosterol-α-d-glucose and *myo*-inositol are reported in the aerial roots [18]. Synephrine, cyanuric acid, adonitol, azelaic acid, butedioic acid, heneicosanyl oleate, *α*-amyrin acetate, lupeol, lanostadienyl glucosyl cetoleate, and bengalensisteroic acid ester have been identified in the methanolic bark extract [62,63]. The leaves contain furanocoumarin (psoralen, rhein, and bergapten) and quinone [61].

## 3. Neuroprotective Effect of *Ficus religiosa*

The neuroprotective effect of *F. religiosa* is summarized in Table 1.

### 3.1. Leaves

The methanolic leaf extract of *F. religiosa* (5–200 µg/mL) exhibits an anti-inflammatory response in LPS-stimulated microglia (BV-2 cells, a mouse microglia cell line) by inhibiting the production of proinflammatory cytokines, such as tumor necrosis factor alpha (TNF-α), interleukin beta (IL-1β) and IL-6, inflammatory mediators and nitric oxide (NO), by downregulating several signaling pathways, such as the p38 mitogen-activated protein kinase (MAPK), extracellular signal-regulated kinase (ERK), and c-Jun N-terminal kinase (JNK) ones. Additionally, the extract also suppresses the activation of nuclear factor kappa B (NF-κB), which strongly supports the neuroprotective role of *F. religiosa* in several NDDs [36].

The neuroprotective effect of the methanolic leaf extract of *F. religiosa* (MEFR) was studied on aluminum chloride (AlCl_3_)-induced neurotoxicity in rats. Aluminum can cross the blood–brain barrier (BBB) through receptor-mediated transfer where it disturbs the oxidative state of the brain and causes neuronal cell death [64,65]. Histological studies have revealed maximum neurodegeneration in the hippocampal CA3 region of the AlCl_3_-treated group. This region has a key role in memory, susceptibility to seizures, and neurodegeneration [66]. However, the group treated with MEFR (200 and 300 mg/kg body weight) presented a significant improvement in the number and quality of neurons [67].

The neuroprotective effect of the petroleum ether (PE) extract of *F. religiosa* leaves (PEFR) was studied on a 3-nitropropionic acid (3-NP)-induced HD mouse model [68]. Systemic intraperitoneal (i.p.) administration of 3-NP caused striatum neuronal degeneration as seen in HD [69]. Daily oral administration of the PEFR (400 mg/kg body weight) significantly enhanced cognitive and motor activities compared to the untreated group. Biochemical investigations showed that the extract reduced the levels of oxidative stress and inhibited the acetylcholine esterase (AChE) activity. However, significant results were not observed at lower doses of the PEFR. The PEFR was safe up to the dose of 4000 mg/kg. These findings regarding the neuroprotective action suggest that *F. religiosa* could be used as an effective therapeutic agent in the management of NDDs. The same group of researchers also evaluated the anti-Parkinson’s activity of the PEFR in haloperidol- and 6-hydroxydopamine (6-OHDA)-induced experimental rat models [68]. A significant reduction in catalepsy induced by haloperidol was observed in the group pretreated with the PEFR indicating that the extract can protect dopaminergic neurotransmission in the striatum. A substantial increase in locomotor activity was observed at 200 and 400 mg/kg. However, the extract was able to reduce levels of malondialdehyde (MDA) and increase the levels of catalase (CAT), glutathione (GSH), and superoxide dismutase (SOD) at a high dose (400 mg/kg body weight) only, suggesting the antioxidant effect of the extract in the brain of 6-OHDA-treated animals. 

The anti-amnesic and nootropic properties of the ethanolic extract of *F. religiosa* leaves (100 mg/kg body weight) were observed in amnesia and hypoxia induced by scopolamine and sodium nitrite in rodents. The results were comparable to the positive controls Piracetam (200 mg/kg) and Mentat (100 mg/kg) [70]. However, contradictory results were observed [71] after oral supplementation of the *F. religiosa* leaf extract in healthy mice where the extract considerably decreased the neuromuscular performance and object recognition ability of male mice only. The conflicting results were, perhaps, due to the extract supplementation to healthy animals as compared to disease models, signifying variable effects of the *F. religiosa* leaf extract in rodents with different health conditions [71].

In addition, the leaves of *F. religiosa* were also evaluated for the anticonvulsant effect [72], but the hydroethanolic extract failed to exhibit a protective effect in pentylenetetrazol (PTZ)- and maximal electroshock (MES)-induced mouse models [73]. 

### 3.2. Root

The saponin-rich fraction (SRF) of the hydroethanolic extract of *F. religiosa* roots (1, 2, and 4 mg/kg) displayed an anticonvulsant effect in mouse models of convulsions [74]. The study was further extended to study the effect of the SRF on cognitive decline and associated depression in a PTZ kindling mouse model of epilepsy [75]. The extract showed marked neuroprotection in the tail suspension test. The SRF considerably raised the monoamine levels and altered the levels of neurotransmitters (noradrenaline, serotonin, γ-aminobutyric acid, dopamine) in the brain. Oral administration of the aqueous root extract of *F. religiosa* (25, 50, and 100 mg/kg) exhibited a dose-dependent and anticonvulsant effect against strychnine- and PTZ-induced seizures. The researchers proposed the involvement of zinc and magnesium present in the extract in the anticonvulsant activity [76].

### 3.3. Fruit

A very high amount of serotonin is known to be present in the fruits of *F. religiosa* [77]. The role of serotonergic neurotransmission in the protection from seizures by modifying various GABAergic and glutamatergic activities is well-documented [78] and the reductions in brain serotonin levels lead to increased susceptibility to seizures [79]. Anticonvulsant studies with the methanolic fruit extract of *F. religiosa* (25, 50, and 100 mg/kg) showed substantial dose-dependent protection in picrotoxin- and MES-induced convulsion mouse models, with the activity similar (at 100 mg/kg) to that observed in the diazepam-treated group. However, PTZ-induced seizures were not inhibited by the extract [42]. 

The flavonoid-rich ethyl acetate fraction of the *F. religiosa* fruit extract (1, 2, and 4 mg/kg i.p.) was used along with a subeffective dose of phenytoin (15 mg/kg) in a PTZ-kindled mouse model. The extract completely suppressed the seizures and reduced the oxidative stress in the brain tissue by decreasing the levels of MDA and increasing the CAT and GSH levels. The extract also decreased the activity of AChE which is responsible for its memory-enhancing effect [75]. 

The methanolic fruit extract of *F. religiosa* (10, 50, and 100 mg/kg i.p.) displayed a dose-dependent anti-amnesic effect in a scopolamine-induced amnesia model of mice [80]. Additionally, inhibition of the anti-amnesic effect of the extract by cyproheptadine (a serotonin antagonist) demonstrated the association of serotonergic pathways for memory improvement by the methanolic extract.

### 3.4. Bark

Numerous plants were screened for AChE inhibitory activity in vitro. Among the screened plants, the methanolic extract of the *F. religiosa* stem bark extract (100–400 µg/mL) displayed the most potent AChE inhibitory activity with an IC_50_ value of 73.69 µg/mL [81].

## 4. Neuroprotective Effect of *Ficus benghalensis*

The neuroprotective effect of *F. benghalensis* extract is summarized in Table 1.

### 4.1. Leaves

Scarce literature is available on the neuroprotective action of *F. benghalensis.* The methanolic leaf extract of *F. benghalensis* (200 and 400 mg/kg p.o.) was evaluated for neuroprotective effects against alloxan-induced diabetic neuropathy in rats [82]. The treated animals had better motor coordination in response to stimuli as compared to the disease control group.

### 4.2. Bark

The methanolic *F. benghalensis* bark extract (100, 200, and 300 mg/kg i.p.) displayed a positive anti-amnesic, anti-anxiolytic, and antidepressant effects in a scopolamine-induced behavioral animal model [62]. The phytocompounds isolated from the extract were identified by GC–MS and might interact with glutamatergic, serotonergic, cholinergic, and GABAergic systems in the brain for memory-improving, anxiety-reducing, and depression-resolving activities observed in the study [62]. The aqueous bark extract (150 and 300 mg/kg body weight) had cognitive enhancement activity in scopolamine-induced amnesia in both old and young mice without any toxicity (up to 5 g/kg) [83].

### 4.3. Root

Oral administration of the aqueous root extract (200 mg/kg body weight) displayed anxiolytic, memory-enhancing, muscle-relaxant, and seizure-modifying effects without any toxicity in mice [84]. The phytochemicals present in the extract were suggested to affect muscarinic receptors in the brain.

## 5. Neuroprotection by Phytochemicals

A vast library of interesting chemicals has been identified in *F. religiosa* and *F. benghalensis* (Figure 1).

### 5.1. Amyrin

Higher concentration of β-amyrin is present in the *B. ceiba* leaf extract, which was reported to ameliorate various biochemical parameters (CAT, MDA, AChE) and cognitive functions in rats with scopolamine-induced amnesia [85]. The memory-enhancing effects of α- or β-amyrin from the mouse model of scopolamine-induced cognitive impairment involved the activation of extracellular signal-regulated kinase (ERK) and inhibition of glycogen synthase kinase (GSK-3β) in the hippocampus [86]. GSK-3β has a role in tau phosphorylation, which ultimately causes their detachment from microtubules and formation of aggregates [87]; hence, inhibition of GSK-3β is of significance in NDDs. The increased activity of ERK in response to oxidative stress and abnormal phosphorylation has been observed in AD [88]. In addition, both ERK and GSK-3β are known to play an important role in synaptic plasticity and memory processes [89,90]. Moreover, amyrin (25 and 50 mg/kg p.o.) also exhibited anticonvulsant activity by increasing the latency time to 75% and 101% at two doses, probably by inhibiting the protein kinase C (PKC) pathway and by increasing taurine (116% and 76%) and tyrosine (135% and 110%) in the basal ganglia and hippocampus, respectively, and decreasing glutamate (68%), aspartate (65%), and GABA (62%) in the basal ganglia [91]. The PCK pathway negatively regulated the expression of the GABA_A_ receptors by affecting the ion channel function and receptor trafficking [92]. Additionally, amyrin exerted potent anxiolytic and antidepressant effects through the inhibition of monoamine oxidase (MAO) and elevating the GABA levels in the hippocampus [93]. MAO-B is known to have a key role in ROS generation, and its inhibitors (selegiline and rasagiline) are used in PD and AD treatment [94]. Hence, amyrin exerts neuroprotection mainly through enhancing the antioxidant pathway.

### 5.2. Azelaic Acid

Azelaic acid’s neuroprotective potential was evaluated in a rotenone-induced PD model (80 mg/kg p.o.) where a significant reversal in posture, muscular rigidity, and catalepsy was observed after the treatments. Additionally, synergistic effects of azelaic acid with levodopa and carbidopa (100 mg/kg + 25 mg/kg p.o.) [95] were revealed, indicating its promising role in treating PD. Azelaic acid has also been identified as a potential biomarker in urine for personalized healthcare in AD diagnosis [96] as its level correlates negatively with Aβ_42_ in cerebrospinal fluid (CSF) and positively with the CSF tau levels. It is also indicative of oxidative damage in the brain of AD patients that may account for changes in brain functions.

### 5.3. Bergapten (5-Methoxypsoralen)

Pieces of evidence support the neuropharmacological effects of bergapten in AD and depression [97,98]. The compound inhibits the AChE and butyrylcholinesterase (BChE) activities in vitro and in silico [99,100,101]. Moreover, bergapten (25 and 50 mg/kg body weight) also improved memory in a scopolamine-induced amnesia model and the cholinergic levels in the hippocampus and the prefrontal cortex without improving motor coordination and locomotor activity [102]. Interestingly, memory improvements were observed after acute and sub-chronic administrations of bergapten, which was independent of the AChE activity and dependent solely on the antioxidant activity of the compound [102]. Additionally, it also exhibited antidepressant effects by inhibiting MAO [103]. Bergapten (25, 50, and 100 mg/kg) reversed the paclitaxel-induced neuropathic pain by restoring the levels of oxidative stress markers (GSH, GST, iNOS, LPO) and downregulating the expression of inflammatory mediators (COX-2, TNF-α, NF-κB) [104]. Briefly, neuroprotection by bergapten is exerted mainly through its antioxidant and anti-inflammatory mechanisms. Additionally, higher bioavailability and the potential to cross the BBB make it a favorable candidate for treating neurological diseases [105,106].

### 5.4. Eudesmol

β-Eudesmol is a major phytocompound in *Atractylodes lancea* rhizome extracts, which induces neurite extensions in PC-12 cells at 100 and 150 µM concentrations by promoting transient phosphorylation of MAPKs (ERK1 and 2). P_38_-MAPK is primarily activated by inflammatory cytokines, playing a critical role in neuronal functions [107]. In addition, β-eudesmol also promotes inositol phosphatase accumulation and encourages the activation of phosphoinositide phospholipase C (PLC), which has a key role in nerve growth factor (NGF)-induced differentiation of neurons [108]. Besides, β-eudesmol (150 µM) induced phosphorylation of the cAMP-responsive element-binding protein (CREB), which is a critical regulator of neuronal plasticity and neuroprotection, in rat pheochromocytoma cells (PC-12) [109]. 

For the treatment of neurological disorders, low-molecular-weight compounds which can easily cross the BBB are preferred. Being a small molecule, β-eudesmol may prove an encouraging lead compound for studying neuronal functions. In short, β-eudesmol displays a distinctive effect on the nervous system by inducing various pathways that are critical for neuronal growth, plasticity, and protection. Studies on the efficacy of β-eudesmol to cross the BBB are in progress [110].

### 5.5. Eugenol

In traumatic brain injury (TBI) rats, pretreatment with eugenol (25, 50, and 100 mg/kg/day p.o., seven consecutive days) ameliorated the neurochemical and behavioral symptoms. Eugenol decreased lipid peroxidation and improved memory and motor activities in the treated group [111]. Eugenol pretreatment (50 and 100 mg/kg) also mitigated the cerebral ischemia/reperfusion (I/R) damage by inducing autophagy activities through the AMPK/mTOR/P70S6K signaling pathway [112]. AMPK and mTOR are the regulators of autophagy, which works through Unc-51-like kinase 1 (ULK1) activation [113]. Activation of AMPK and inhibition of mTOR endorses the autophagy activities. Additionally, P70S6K (ribosomal protein S6 kinase beta-1) is a downstream kinase of mTOR, whose suppression may promote autophagy activities [114].

In another study, eugenol treatment (10 mg/kg bw for 5 weeks) improved the gait in acrylamide-induced neuropathic rats and restored the levels of antioxidant enzymes and dopamine in the brain [115]. Furthermore, cotreatment with eugenol (6 mg/g) in aluminum-induced toxicity in rat brains reduced the AChE, TNF-α, and caspase-3 expression [116]. Additionally, it promoted neurogenesis in the hippocampus by increasing the expression of the metallothionein gene (MT-III) and restored the levels of brain-derived neurotrophic factor (BDNF) and serotonin in the brain [117]. Since BDNF is vital for the preservation of cortical neurons, its initial loss would lead to short-term memory decline in AD [118]. BDNF expression could be induced by NGF through phosphorylating the CREB in the ERK/AKT signaling pathway [119]. Eugenol (0.1, 1, and 10 mg/kg orally) also displayed neuroprotective properties in a hydroxydopamine-induced PD model [120]. In an in vitro ThT (thioflavin T) assay, eugenol (3 mM) also suppressed amyloid formation by delaying the conversion to the β-sheet form [121]. 

In another interesting study, eugenol and its analogs were reported to interact with vanilloid receptors in the olfactory bulb and displayed a positive effect on memory [122]. Such a possibility can be explored for the treatment of AD and PD. 

Eugenol appears to be a wonder molecule because of its ability to cross the BBB [116] and displays the neuroprotective effect through neurogenesis, antioxidant, anti-amyloid, and antiapoptotic ability by affecting multiple pathways. This multitarget approach expands the application of eugenol to multiple neurological diseases. 

### 5.6. Kaempferol

Kaempferol is a multipotential neuroprotective agent that affects various pathways in NDDs [123]. In neurological diseases, the antioxidant activity of kaempferol inhibits various metalloproteases (MMP-2, MMP-3, MMP-9) and protects the BDNF modulations responsible for neuronal plasticity. It is known for enhancing cognitive performance in animal models by displaying anti-AChE activity [124,125]. Kaempferol and its degradation products inhibit Aβ oligomerization and plaque formation by interacting with the Aβ protein without involving the Lys_16_ and Lys_18_ residues [126] and restores the levels of oxidative stress markers (SOD, glutathione, ROS) both in vitro and in vivo [127]. It also displays its neuroprotective potential at 20 mg/kg dose by suppressing microglial activation by inhibiting the NF-κB, MAPK, AKT, and toll-like receptor 4 (TLR4) pathways in an LPS-induced striatum injury mouse model [128]. 

In a rotenone-induced PD rat model, kaempferol (10 and 20 mg/kg) increased the SOD and glutathione peroxidase (GSH-P_X_) levels and decreased lipid peroxidation. It neutralized ROS by blocking apoptosis through the JNK/MAPK pathway [129]. It also reduced the expression of proinflammatory cytokines, COX-2, and the high mobility group box 1 (HMGB1)/TLR4 inflammatory pathway [128]. As a result, motor coordination and dopamine levels increased in the PD model [130]. In summary, kaempferol displays multitarget properties (antioxidant, anti-inflammatory, anti-amyloid, antiapoptotic, and modulating enzymes involved in neurotransmission) for neuroprotection. Moreover, due to its ability to cross the BBB [131], it can prove beneficial in the treatment of NDDs.

### 5.7. Lanosterol

In vitro and in silico studies have indicated that lanosterol can suppress the buildup of misfolded protein aggregations/sequestosomes [132,133] by promoting autophagy activities [134]. Lanosterol (0.5 mM) also protected dopaminergic neurons from 1-methyl-4-phenylpyridinium (MPP+)-induced cell death in a PD cellular model by inducing mild mitochondrial depolarization and promoting autophagy activities. The observed reallocation of lanosterol synthase to mitochondria suggested that lanosterol might exert its survival effect by regulating mitochondrial functions [135]. In addition, reduced levels of lanosterol were observed in the striatum and ventral midbrain regions from 1-methyl-4-phenyl-1,2,3,6-tetrahydropyridine (MPTP)-treated mice (PD model), suggesting a potential role in the cholesterol metabolism of NDD pathogenesis. In short, lanosterol promotes autophagy activities by sequestering misfolded proteins.

### 5.8. Leucoanthocyanins

Leucoanthocyanins are the intermediates in the synthesis of anthocyanins, which, in turn, are reported to contain multitarget neuroprotective properties in animal and in vitro models of NDDs [136,137,138,139] by reducing oxidative stress and modulating anti-inflammation [140,141] through the phosphoinositide 3-kinase (PI3K)/protein kinase B (Akt)/nuclear factor erythroid 2-related factor 2 (Nrf2)/heme oxygenase-1 (HO-1) pathway [55] and cyclooxygenase-2/microsomal prostaglandin E synthase-1 (COX-2/mPGES-1) [142]. The phosphorylation of PI3K activates Akt, which, in turn, disables GSK-3β by phosphorylation (p-GSK-3β). The latter helps in the translocation of Nrf2 to the nucleus, where it enhances the expression of antioxidant genes (including HO-1) [143,144,145]. Furthermore, anthocyanins also promote autophagy by upregulating expression of autophagy-related proteins through the AMP-activated protein kinase and mammalian target of rapamycin (AMPK–mTOR) signal pathway [146]. Evidence also suggests the BBB-crossing capability of anthocyanins [147] which would be an advantage in treating NDDs. In summary, leucoanthocyanins exert their neuroprotection through antioxidant, antiapoptotic, and anti-inflammatory pathways.

### 5.9. Limonene

Limonene (10 µg/mL) presents a neuroprotective potential against exposed primary cortical neurons to Aβ_1–42_ oligomers (in vitro model of AD) by mitigating ROS generation and the potassium channel (K_V_3.4) hyperfunction [148]. In AD, enhanced ROS eventually cause Aβ_1–42_-induced upregulation of K_V_3.4. In addition, Ca^2+^-induced ROS production activates K_V_3.4 through the NF-κB pathway [149]. Limonene present in the essential oil mixture (MO: 1% and 3%) has been known to revert cognitive deficits in the scopolamine-induced amnesia rat model by alleviating the oxidative stress markers (MDA, SOD, GSH) and inhibiting AChE (24.9%) and BchE (69.1%; IC_50_, 1.096 ± 0.043 µg/mL) activities. Administration of limonene (5, 25, and 50 mg/kg for 1 week) also significantly increases GABA, a key hypothalamic neurotransmitter, in the rat brain. This increased activity inhibits the release of corticosterone from the hypothalamic–pituitary–adrenal (HPA) axis under stress conditions, thus playing a vital role as an antistress agent [150]. Molecular docking models have revealed the van der Waals interaction between limonene and active side residues (Ser_198_, His_438_, Leu_286_, Val_288_, Phe_329_) of BchE [151]. Different plant extracts with high limonene concentrations also exhibit neuroprotection by endorsing potential anti-AchE, anti-inflammatory, and antioxidant activities [152,153,154].

### 5.10. Linalool

Linalool has shown a protective effect in various neurodegenerative models and is reported to have the ability to cross the BBB [155]. In in vitro experiments, linalool (10 µg/mL) protected PC-12 cells from Aβ exposure by reducing ROS and inhibiting the activity of proapoptotic caspase-3 [156]. Linalool (50 and 100 mg/kg/day for 21 days) also significantly suppressed Aβ-induced ROS, oxidative stresses, and inflammatory responses in an AD fly model [157] without altering the amount of Aβ in the brain. Alterations in the hippocampal phospholipid profiles in ischemic animals were mitigated by linalool, which helped to maintain the phospholipid homeostasis, hence recovering brain functions [158]. The oral administration of linalool (25 mg/kg for 3 months) in a triple transgenic AD mouse model (3xTg-AD) restored memory in the treated animals via reducing β-amyloidosis, astrogliosis, and tauopathy besides reducing proinflammatory markers (MAPK, inducible oxide nitric synthase (iNOS), COX-2, IL-1β) [159]. Linalool (100 µM) is known to modulate glutamatergic neurotransmission by interacting with NMDA receptors [160]. Furthermore, it proved to be neuroprotective in a glutamate-induced oxidative stress in vitro model (HT-22 cells) by reducing ROS, calcium production, and LPO levels [161]. Linalool (162, 324, 648 μM) exerts protective effects in LPS-induced BV2 microglial cells by Nrf2 activation [162] and in mice with Aβ-induced cognitive deficits by restoring the levels of oxidative stress-related enzymes, suppressing caspase-3, and upregulating the Nrf2 and HO-1 expression [163]. Together, these findings suggest that linalool could be used for the development of NDD drugs as it can cross the BBB and provide neuroprotection through anti-inflammatory, antioxidant, and antiapoptotic properties.

### 5.11. Lupeol

Lupeol (triterpenoid) is reported for its antioxidative, anti-inflammatory, and neuroprotective activities in a variety of animal models [164]. Lupeol competitively inhibited β-secretase (BACE-1) (IC_50_, 5.12 μmol/L) with a low inhibition constant (K_i_ 1.43 μmol/L), indicating better affinity. Molecular docking revealed the formation of hydrogen bonds between the hydroxyl group of lupeol and the Asp_32_ and Ser_35_ of BACE-1 [165], upregulating the expression of proinflammatory cytokines and interleukins (TNF-α, IL-6, IL-1β) at 25, 50, and 100 mg/kg p.o. concentrations in acetic acid-induced writhing, the formalin test, carrageenan-induced hyperalgesia, and a postoperative pain model [166], activating the Nrf2/HO-1 pathway and improving cognitive functions (at a 50 mg/kg dose) in Aβ-induced oxidative stress in mice [167]. Additionally, lupeol (50 mg/kg dose p.o.) also inhibited apoptotic signaling molecules (caspase-3, BCL2-associated X (Bax), cytochrome c) and repressed astrocytes/microglia activation in the cortex and the hippocampus of a TBI mouse model [168]. Lupeol (0.1 µM) downregulated the anti-inflammatory responses of TNF-α, iNOS, and NOD-like receptor pyrin domain-containing protein 3 (NLRP3) and upregulated the arginase, IL-6, neurotrophin (glia-derived neurotrophic factor (GDNF)), and sonic hedgehog–Gli (SHH–GLI) signaling [169] in cerebellar cultures and induced neuroprotection. Additionally, lupeol displayed a superior ADMET (absorption, distribution, metabolism, excretion, toxicity) profile and proved nontoxic, noncarcinogenic, biodegradation-resistant, as well as low inhibition by cytochrome P450 (CYP_450_). Most importantly, it can cross the BBB easily [170].

Hence, lupeol seems to be a potent candidate for NDD drug discovery as it has an acceptable ADMET profile and the multitarget neuroprotective approach (inhibiting neuroinflammation, reducing oxidative stress, repressing apoptosis and microglial activations). 

### 5.12. Myo-Inositol (Vitamin B8)

Even though no direct study has been conducted to investigate the effect of *myo*-Inositol on the NDD model systematically, it displays a neuroprotective role in ischemic stroke injury in animals exposed to tobacco smoke and in streptozotocin-induced mice by increasing the motor functions after the stroke using in situ brain perfusion and the acute brain slice method at 0.1 μCi/mL [171,172]. In addition, myo-inositol (30 mg/kg for 28 days) ameliorated spatial learning and memory deficits by attenuating cell loss in the hippocampus in a kainic acid-induced epilepsy rat model [173] by a multitarget approach including preserving neuronal circuits and activation of GABA signaling. *Myo*-inositol also presented the anticonvulsive property in thiosemicarbazide models of seizures by increasing the latent time and decreasing the severity of the seizure [174]. 

Interestingly, *myo*-inositol has been considered a noninvasive early marker for assessing various asymptomatic AD stages in comparison with magnetic resonance spectroscopy (MRS) [175]. Increased *myo*-inositol/creatine levels have been observed in healthy apolipoprotein E E4 genotype (APOE ε4) carriers with normal CSF Aβ42 levels in comparison to ε4 non-carriers, signifying the importance of *myo*-inositol levels in assessing AD before a noticeable amyloid pathology [176]. Reports suggest that myo-inositol can also be transported across the BBB by simple diffusion as well as through a stereospecific transporter [177]. 

### 5.13. Myricetin

Myricetin is a lipophilic compound with the ability to cross biological membranes, including the BBB [178]. It is reported to target AD by inhibiting multiple pathways, such as neuroinflammation, autophagy, oxidative stress, chelation, anti-AchE, and Aβ depositions [179]. Myricetin (5 μM) averted Aβ_1–42_ oligomer-induced neurotoxicity in human neuroblastoma cells (SH-SY5Y) by exerting antioxidant effects on the cell membranes and mitochondria. Myricetin also restored mitochondrial dysfunctions by decreasing ROS, increasing the expression of manganese superoxide dismutase (Mn-SOD) and ATP generation [180]. The JNK/stress-activated protein kinase (SAPK) pathway is activated by oxidative stress and Aβ. Enhanced BACE-1 levels lead to an increase in Aβ levels [181], which eventually activates the JNK/SAPK pathway, resulting in a vicious cycle of the NDD. Furthermore, oxidative stress could also endorse serine/threonine protein phosphatase (PP2A) inhibition, promoting tau phosphorylation [182] and damaging the mitochondria [183]. Myricetin inhibited the BACE-1 activity (IC_50_, 2.8 μM) to cleave the amyloid precursor protein (APP), increase α-secretase, and decrease Aβ production/oligomerization [184,185]. Additionally, myricetin also interferes with the NF-κB and AMPK/SIRT1 signaling pathway and reduces the levels of inflammatory mediators (IL, TNF-α, iNOS, COX-2) in the brain [186]. Myricetin (1 and 10 μM) also aids in the removal of abnormal Aβ and tau through autophagy activation by inhibiting phosphorylation of mammalian targets of rapamycin (mTOR) in primary neuron cultures [187]. Interestingly, myricetin (25 μM) can regulate the levels of metal ions in the brain to reduce their interactions with Aβ and disassemble the formation of the metal–Aβ complex [188]. Since it was previously reported that Fe^2+^ could activate microglia by increasing neuroinflammation, the complexation of myricetin with Fe^2+^ could reduce the inflammatory processes and inhibit the expression of transferrin receptor I (TrP1), thus lowering the iron levels [6]. Lastly, myricetin also improves learning and cognition through the anti-AchE activity. Some proinflammatory cytokines (IL-1) activate AchE causing the Ach levels to decrease in the brain, affecting memory. Hence, the anti-inflammatory activity of myricetin also improves memory [189,190]. 

### 5.14. Pinene

Pinene, a monoterpene, can cross the BBB and affect multiple neurotransmitter systems, such as adrenergic, cholinergic, dopaminergic, GABAergic, serotoninergic, and noradrenergic functions in the brain [191]. According to a cell-based study, pretreatment of α-pinene (10 and 25 µM) in PC-12 cells inhibited ROS by increasing the expression of antioxidant enzymes and reducing apoptosis by decreasing the caspase-3 activity [192]. α-Pinene administration (50 mg/kg i.p.) in the Aβ-induced rat model reduced neuroinflammation by overturning the TNF-α/NF-κB pathway and improved memory and learning. Moreover, α-pinene also upregulated the expression of both the nicotinic acetylcholine receptor (nAChR) α7 subunit and BDNF [193], which play a role in the survival and maintenance of neurons. Additionally, α-pinene (100 mg/kg i.p.) restored the levels of antioxidant enzymes (SOD, CAT, GPX) and reduced NO, IL-6, and MDA in the brain of focal ischemic stroke model rats [194]. Furthermore, α-pinene (100 mg/kg i.p.) downregulated Bax with a corresponding upregulation of Bcl-2 expression, resulting in suppression of apoptosis in a rat model of cerebral ischemia-reperfusion [195]. 

In a mouse model of memory impairment, α-pinene upregulated the expression of AchE, muscarinic receptors, and antioxidant transcription factors in the hippocampus, improving spatial recognition and memory [196]. In summary, the antioxidant, anti-inflammatory, and antiapoptotic properties of α-pinene would help treat NDDs.

### 5.15. Psoralen

Psoralen is a natural furanocoumarin with in vitro competitive inhibitory activity against AchE (IC_50_, 370 μg/mL) [197]. A molecular docking study revealed a stable AchE–psoralen complex with π–π stacking (Tyr_334_) and hydrogen bonding (Gly_119_ and Gly_118_) interactions [197]. Psoralen also displayed inhibitory activities towards MAO-A (IC_50_, 15.2 µM; noncompetitive inhibition) and MAO-B (IC_50_, 61.8 µM; competitive inhibition) in the rat brain [198]. Since MAO is an important enzyme in maintaining levels of monoamine neurotransmitters in the brain, its reduced expression results in decreased Aβ depositions and oxidative stress [199]. In another study, psoralen and iso-psoralen-rich extracts (0.1 and 0.3 mg/kg) of *Psoraleae fructus* improved amnesia in scopolamine-induced rats, apparently by AchE inhibition (IC_50_, 1.12 mM) and activation of cholinergic neuronal functions [200]. In summary, even though moderate enzyme inhibitory activity has been observed with psoralen, it could serve as a lead molecule to synthesize other potential analogs to treat NDDs.

### 5.16. Quercetin

Quercetin (20 μM) decreases Aβ production in primary neuron cultures by inhibiting BACE-1 (IC_50_, 5.4 μM). A molecular docking study identified the interaction of quercetin with the Asp_32_ of BACE-1 [201], which resulted in cognitive improvement in the animal model of NDD [202]. Additionally, quercetin (20 and 40 μM) exerted neuroprotective effects by protecting proteins and lipids from oxidation in case of Aβ_1–42_-induced toxicity [203]. Quercetin (10 and 30 μM) also safeguarded mitochondria by reducing the production/accumulation of ROS, NO, increasing GSH, decreasing overexpression of proinflammatory cytokines, and reducing dopaminergic degeneration of neurons in MN9D cells (mouse dopaminergic cell line) and a MitoPark PD animal model [204]. A positive effect of quercetin supplementation was also observed in a 3-NP-induced Huntington’s disease model (HD). Quercetin (25 mg/kg p.o. for 21 days) restored the levels of ATP, CAT, and SOD, relieving mitochondrial oxidative stress. Histopathological studies reported diminished striatum astrogliosis and pyknotic nuclei in the HD model [205]. Quercetin supplementation seemed to mitigate the biochemical and neurochemical changes in the rat brain by altering/inhibiting inflammatory activities from NDDs [206]. The targeted downstream pathways of quercetin [207] for neuroprotection are as follows: paraoxonase 2 (PON2) [208,209], Nrf2-ARE; phosphoinositide 3 kinase (PI3K) [210], JNK/ERK [211], TNF-α [212], peroxisome proliferator-activated receptor gamma coactivator 1-alpha (PGC-1α), and SIRT1 [213], CREB [214], MAPK [215], NF-κB [210], and AMPK [216]. 

In essence, quercetin presents a multitarget tactic (anti-inflammatory action, antioxidant action, and enzyme inhibition) to ameliorate the symptoms of NDDs.

### 5.17. Rhein

Rhein (12 mg/kg) displayed antioxidative and neuroprotective effects in controlled cortical impact (CCI) rats by reducing the MDA levels and increasing SOD, CAT, and GSH [163], thereby protecting the brain from oxidative damage. It also protected the BBB through the NADPH oxidase/ROS/ERK/MMP-9 signaling pathway [217]. Additionally, in an I/R rat model, rhein (50 and 100 mg/kg/day for 3 days) restored the levels of antioxidant markers, enhanced Bcl-2, and decreased the levels of Bax and caspase-3 [218]. Rhein (100 mg/kg i.p.) also exerted an anti-inflammatory response by downregulating the inflammatory cytokines in the cortex of animals with TBI [219]. Additionally, rhein (10 mg/kg i.v.) improved the cognitive decline in an APP/PS1 mouse model of AD by activating the SIRT1/PGC-1*α* pathway, improving oxidative stress by regulating mitochondrial biogenesis [220]. It also exerted an anti-inflammatory response by subsiding the levels of TNF-*α* and IL-1β in the hippocampus and an antioxidant property by decreasing MDA, increasing GSH, the GSH/GSSG ratio, CAT, and GSH-Px in the same model [220]. The pharmacokinetics of rhein disclose that it can be easily transported by body fluids and can pass through damaged BBB which is very useful in case of brain injury [221]. In summary, rhein could be an important compound for neuronal protection through antioxidant, anti-inflammatory, and antiapoptotic properties.

### 5.18. Rutin

Rutin has emerged as an important pharmacological flavonoid in various NDDs [222] as it can easily cross the BBB [223]. Rutin improves memory by reducing Aβ oligomerization, oxidative stress, neurotoxicity, and neuroinflammation in several AD animal models [224,225,226,227]. The anti-amyloidogenic property could be due to the destabilization of Aβ by direct interaction of the aromatic ring of rutin with the hydrophobic β-sheet of amyloid aggregates. Rutin (100 μL rutin suspension/per 10 g bw for 30 days) decreased the tau levels by regulating phosphorylation through increased PP2A activities. It also reduced inflammation by downregulating NF-κB besides securing neuronal morphology and improving cognition through synapse preservation in the brain of the Tau-P301S mouse model [228]. In addition to the AD models, rutin (25 mg/kg bw orally for 3 weeks) protected dopaminergic neurons by reducing oxidative stress and apoptosis in the 6-OHDA-induced rat model of PD [229,230]. It also improved HD symptoms by activating autophagy and the insulin/insulin-like growth factor I (IGF-1) pathway at 15–120 μM concentrations [231]. APP phosphorylation is suppressed by insulin/IGF-I in vitro which favors the non-amyloidogenic pathway [232]. This pathway regulates neurogenesis and is known to improve neuronal survival, learning, and memory through the PI3K/Akt and ERK pathway [233,234,235].

In brief, rutin provides neuroprotection through its antioxidant, anti-inflammatory, anti-amyloid, and autophagic activation properties.

### 5.19. Stigmasterol

Stigmasterol, a phytosterol, demonstrated its neuroprotective effects through multitarget approach in vitro and in vivo. Studies revealed that phytosterols can cross the BBB through the scavenger receptor class B member 1 (SR-BI)-dependent pathway and via apolipoprotein E (ApoE) [236], improving the cognition [237]. It reduced amyloid plaque formations [238], inhibited the AchE activity (IC_50_, 644 μM) in vitro [239], and decreased the elevated ROS [240]. Stigmasterol (1 μM) displayed its neuroprotective effects in hydrogen peroxide-induced oxidative stresses in SH-SY5Y cells by modulating the sirtuin 1-forkhead box O3a (SIRT1-FoxO3a) pathway [241]. SIRT1 activates FoxO3a, which consequently stimulates the production of antioxidant enzymes (SOD, CAT), hence protecting against oxidative stress. Stigmasterol also exhibited anti-inflammatory activity in the IL-1β-treated cells by inhibiting proinflammatory cytokines without affecting the levels of anti-inflammatory cytokines suggesting its role in the NF-κB inflammatory pathway [242]. In a recent report, stigmasterol (10 and 20 μM) inhibited NF-κB and NOD-like receptor thermal protein domain-associated protein 3 (NLRP3) signaling by activating AMPK, thereby reducing the Aβ-induced inflammatory response in BV2 cells [243]. Stigmasterol (3, 10, and 30 mg/kg) also exerted ameliorating effects on the scopolamine-induced memory loss in mice through the cholinergic neurotransmission augmentation by N-methyl-D-aspartate receptor (NMDA) activation [244]. Earlier reports had shown progressive modulatory effects of Ach by binding to muscarinic receptors on NMDA, enabling NMDA receptor-mediated synaptic plasticity and long-term potentiation (LTP), which are implicated in learning and memory [245]. Besides, stigmasterol (10 mg/kg p.o.) has been reported to increase the ERK and CREB phosphorylation in the hippocampus of the scopolamine-induced memory loss model in mice, which could also influence a positive effect on memory and learning [244,246].

### 5.20. Synephrine

Synephrine is a sympathomimetic alkaloid with mild CNS stimulant properties [247]. When the effects of oral supplementation of synephrine (20 mg) were evaluated for cognition and exercise performances during a pre-workout, no significant results were observed on muscular endurance during exercise, but it seemed to improve cognitive functions and mental focus [248]. Synephrine has effective inhibitory properties towards AchE (IC_50_, 226.01 nM) and BchE (IC_50_, 92.22 nM) in vitro [249].

### 5.21. β-Caryophyllene

β-Caryophyllene is a natural sesquiterpene that can easily cross the BBB and exert neuroprotective effects [250]. Additionally, β-caryophyllene (48 mg/kg p.o. for 7 weeks) reduced Aβ deposition in the cerebral cortex and the hippocampus of APP/PS1 mice. It also reduced the levels of COX-2, TNF-α, and IL-β in the cerebral cortex. β-Caryophyllene is a known cannabinoid receptor 2 (CB2) agonist, and the activation of CB2 receptors is beneficial in reducing neuroinflammation by triggering the peroxisome proliferator-activated receptor-γ (PPARγ) pathway [251]. β-Caryophyllene (5 μM) also inhibited the hypoxia-induced neuroinflammatory processes in BV2 cells by mitigating ROS production and proinflammatory cytokines by inhibiting p38MAPK/NF-κB [252]. β-Caryophyllene (24, 72 mg/kg i.p.) prevented neuronal necrosis, downregulated the receptor-interacting protein kinase-1 and -3 (RIPK1, RIPK3), MLKL phosphorylation, HMGB1, TLR4, and proinflammatory cytokines. Thus, β-caryophyllene exerts neuroprotection by inhibiting inflammation and neuronal death in a cerebral I/R injury mouse model [253]. Caryophyllene is generally considered safe (GRAS) by the FDA for use in the food industry [254].

### 5.22. β-Sitosterol

β-Sitosterol displayed robust anti-AchE (55 µg/mL) and anti-BchE (50 µg/mL) activity both in vitro and in silico. The in vivo results confirmed that β-sitosterol acts as a free radical scavenger and can reach the brain and inhibit AchE and BACE [255,256]. Its antioxidant effects (15 µM) were observed in a glucose oxidase (GOX)-induced oxidative stress and lipid peroxidation model of HT22 hippocampal cells through the estrogen receptor (ER)-mediated PI3K/GSK-3β signaling pathway [257]. β-Sitosterol seemed to help the PI3K recruitment to the lipid raft, an important region of the membrane in signal transduction [258]. GSK-3β is an important downstream target of PI3K upregulation which eventually increases the intracellular glutathione, a natural antioxidant [259]. Additionally, β-sitosterol augmented the mitochondrial membrane potential (ΔΨm) and adenosine triphosphate (ATP) by integrating into the mitochondrial membrane [257].

In NDDs, neurons are damaged due to persistent neuroinflammation. In a study, β-sitosterol (8 and 16 µM) displayed anti-inflammatory properties in lipopolysaccharide (LPS)-induced BV2 microglial cells by reducing the expression of proinflammatory factors (IL-6, iNOS, TNF-α, cyclooxygenase-2 (COX-2), an inhibitor of nuclear factor kappa B: IκB, NF-κB, ERK/p38) [260]. In animal models, β-sitosterol exhibited positive effects on learning and memory [255,261,262], and prevented plaque deposition in an amyloid protein precursor/presenilin 1 (APP/PS1) model [255,261,262]. Recently, substantial anxiolytic and antidepressant effects of γ-sitosterol were observed from the strong binding affinity with the human serotonin receptor in a molecular docking study [263]. In short, the neuroprotective effects of β-sitosterol would be due to its antioxidant, anti-inflammatory, anti-amyloid, and enzyme inhibition properties. It seems to be a potential candidate for managing memory-related disorders in NDDs. Pan-assay interference structures (PAINS) are chemical compounds which could give a significant false positive signal in the drug screening processes including redox reactivity, fluorescence of small compounds, and covalent changes of target proteins [264,265]. For instance, curcumin, known as the representative PAINS, has failed more than 120 clinical studies for various diseases due to false activity in vitro and in vivo [266]. Moreover, at least 15 studies have been retracted and dozens more have been corrected since 2009 [267]. As a result, it is imperative for accurate assessment of these compounds with structural alerts and elimination from further steps of the drug discovery process [268]. Medicinal chemistry analysis of PAINS was processed by the SwissADME server [269]. Except for myricetin, quercetin, rhein, and rutin, all the phytocompounds derived from *Ficus* trees were predicted to have no structural alerts or false positive signals (Table 2). 

The summary of neuroprotective mechanisms of the discussed phytocompounds from sacred *Ficus* trees is described in Table 2 and Figure 2.

## 6. Conclusions and Future Directions

Even though plant extracts have been used for centuries for treating a spectrum of diseases in traditional medicine, no scientific validation for their therapeutic effects has been presented. In the modern era, the use of plants for therapeutic purposes was underestimated initially. Recently, after the discovery and use of numerous important drugs from herbal sources [270], such as quinine (an antimalarial drug from the *Cinchona* bark), atropine (anticholinergic medicine from *Belladona*), digoxin (obtained from *Digitalis* to treat cardiac arrhythmia), colchicine (extracted from *Colchicum* for treating gout), and galantamine (AChE inhibitor from *Galanthus* spp.), the interest in plant-based research has been expanding to determine their mechanistic actions behind the therapeutic potentials and has reached a new height. In an extract, a cocktail of various bioactive compounds may exhibit synergistic effects for better therapeutic activity, questioning whether a purified compound would present similar effects. However, an understanding of a purified bioactive compound should be performed first, even if a cocktail of compounds may be needed down the line. In addition, since the pathophysiology of NDDs would be complex, linking multiple cellular events, a multitarget tactic might be a better approach going forward. 

Several bioactive compounds in the sacred *Ficus* species display neuroprotective properties in vitro and in vivo through multiple pathways, such as antioxidant (PI3K, AKT, AMPK, PKC, ERK, HO-1, Nrf2), anti-inflammatory (SIRT1, NF-κB), anti-amyloid (APP, BACE-1), antiapoptotic (Bcl-2, Bax, caspase), and modulating enzymes in neurotransmission (AChE, BChE). Most of the studies were carried out using rodent models of NDDs that unfortunately cannot recapitulate the complete aspects of AD as it is a uniquely human disease [271]. This is one of the reasons why many drugs were unsuccessful in clinical trials. Therefore, performing research using human tissues would give a more human-centric strategy. 

Additionally, the negative results of the anti-Aβ strategy (amyloid cascade hypothesis) in clinical trials on AD patients demonstrate that it is not the only pathogenic factor involved. Pulling down cerebral Aβ can only delay cognitive decline but it cannot stop it, indicating the role of other etiological factors (oligomer cascade hypothesis) in the pathogenesis of AD [272]. Hence, scientists must follow a multitarget approach, leading to the treatment of complex diseases, such as NDDs. 

In conclusion, *Ficus* spp. extracts and bioactive compounds present effective neuroprotective properties (in vivo and in vitro) by modulating several important pathways. The results from in vivo experiments also indicate the nontoxic nature of the extracts/phytocompounds at the doses tested. Yet, they have not been translated into clinical trials. Henceforth, the need to take the research to next level is of great significance in the treatment of NDDs.

## Figures and Tables

**Figure 1 nutrients-14-04731-f001:**
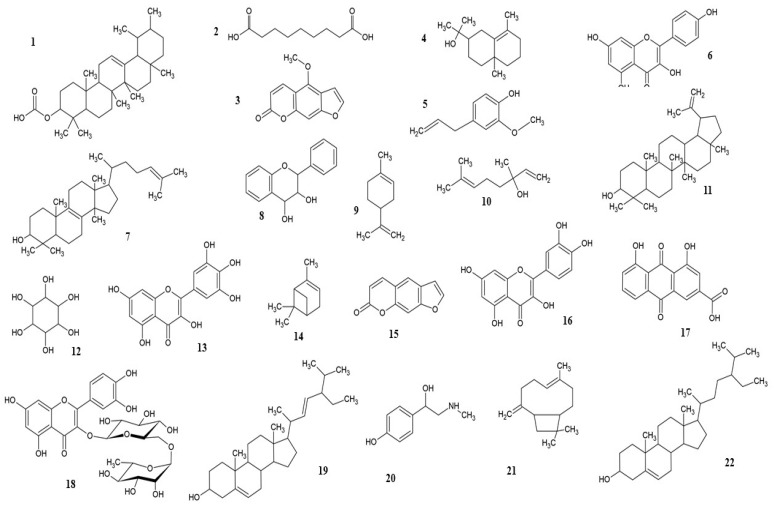
Some important phytocompounds from *F. religiosa* and *F. benghalensis*: **1**, amyrin; **2**, azelaic acid; **3**, bergapten; **4**, eudesmol; **5**, eugenol; **6**, kaempferol; **7**, lanosterol; **8**, leucoanthocyanins; **9**, limonene; **10**, linalool; **11**, lupeol; **12**, myo-inositol; **13**, myricetin; **14**, pinene; **15**, psoralen; **16**, quercetin; **17**, rhein; **18**, rutin; **19**, stigmasterol; **20**, synephrine; **21**, β-caryophyllene; **22**, β-sitosterol.

**Figure 2 nutrients-14-04731-f002:**
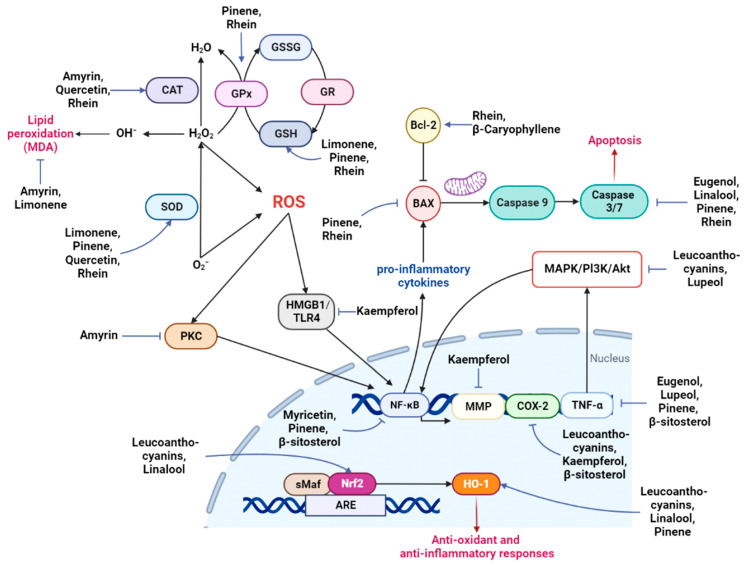
Mechanism of neuroprotection by various important phytochemicals from *F. religiosa* and *F. benghalensis*. Major phytochemicals act as inhibitors against oxidative stress and inflammation pathways. Oxidative stress is modulated by triggering antioxidant enzymes or transcription factors and suppressing ROS-mediated proteins. The cascade of inflammation is hampered by the downregulation of inflammatory transcription factors and caspase-related components. Abbreviations: CAT, catalase; GPx, glutathione peroxidase; GR, glutathione reductase; GSH, glutathione; GSSG, oxidized glutathione; GST, glutathione S-transferase; SOD, superoxide dismutase; ROS, reactive oxygen species; MDA, malondialdehyde; PKC, protein kinase C; HMGB1, high mobility group box 1; TLR4, toll-like receptor 4; Bcl-2, anti-apoptotic B cell lymphoma-2; Bax, Bcl-2-associated X; NF-κB; nuclear factor kappa B; MAPKs, mitogen-activated protein kinases; PI3K, phosphoinositol-3 phosphate; AKT, protein kinase B; MMP, matrix metalloproteinase; COX-2, cyclooxygenase 2; TNF-α, tumor necrosis factor α; sMAF, small musculoaponeurotic fibrosarcoma protein; Nrf2, nuclear E2-related factor 2; ARE, antioxidant response element; HO-1, heme oxygenase-1.

**Table 1 nutrients-14-04731-t001:** Neuroprotective mechanism of *Ficus* extracts.

Name	Plant Part	Extract	Model	Dose (mg/kg)	Action	Ref.
*F. religiosa*	Leaves	Methanolic	BV2 cell lines		Inhibits proinflammatory cytokine production; downregulates MAPK/ERK/JNK/NF-κB; improves the number and quality of neurons	[36,64,65,66,67,68,69,70,71,72,73]
			AlCl3-induced	200 and 300		
		Petroleum ether	3-NP-, 6-OHDA-induced	200 and 400	Anti-AChE; reduces oxidative stress	
		Ethanolic	Scopolamine-, sodium nitrite- induced	100	Anti-amnesic and nootropic	
	Root	Hydroethanolic	PTZ-induced	1, 2, 4	Anticonvulsant	[74,75,76]
		Aqueous	Strychnine-, PTZ-induced	25, 50, 100	Anticonvulsant	
	Fruit	Methanolic	MES-, picrotoxin, scopolamine-induced	25, 50, 100	Antiamnesic, anticonvulsant	[77,78,79,80]
				10, 50, 100		
		Ethyl acetate	PTZ-induced	1, 2, 4	Reduces oxidative stress, anticonvulsant, anti-AChE	
	Bark	Methanolic	In vitro		Anti-AChE	[81]
*F. benghalensis*	Leaves	Methanolic	Alloxan-induced	200 and 400	Improves motor coordination	[82]
	Bark	Methanolic	Scopolamine- induced	100, 200, 300	Anxiolytic and antidepressant	[62,83]
		Aqueous	Scopolamine- induced	150 and 300	Cognitive enhancement	
	Root	Aqueous	PTZ-, MES-induced	100 and 200	Anxiolytic, memory-enhancing, muscle-relaxant, seizure-modifying effect	[84]

**Table 2 nutrients-14-04731-t002:** Neuroprotective mechanism of some important phytocompounds from sacred *Ficus* trees.

Name	Class and MW	BBB Permeability	Model	Dose/Concentration	MOA	Pathways Affected	Medicinal Chemistry (PAINS)	Ref.
Amyrin	Phytosterol 426.72		PTZ-induced seizures	25 and 50 mg/kg	Antioxidant	ERK activation, GSK inhibition, memory enhancement; MAO inhibition; elevation of GABA; inhibits PKC; increases CAT; decreases MDA; inhibits AChE	0 alerts	[85,86,87,88,89,90,91,92,93,94]
Azelaic acid	Dicarboxylic acid 188.22		Rotenone-induced PD model	80 mg/kg		Improves motor functions	0 alerts	[95,96]
Bergapten	Furanocoumarin 216.19	**√**	Scopolamine-induced amnesia; paclitaxel-induced neuropathic pain	25 and 50 mg/kg; 25, 50, and 100 mg/kg	Enzyme inhibition	Inhibits AChE, BchE, and MAO; memory enhancement; anti-depressant	0 alerts	[97,98,99,100,101,102,103,104,105,106]
Eudesmol	Sesquiterpenoids 222.37		PC-12 cells	100 and 150 µM	Neurite extension	Induced neurite extension; MAPK activation; phosphorylation of the CREB	0 alerts	[107,108,109,110]
Eugenol	Polyphenol 164.2	**√**	TBI rats; I/R damage; acrylamide-induced neuropathic rats; aluminum-induced toxicity; hydroxydopamine-induced PD model	25, 50, and 100 mg/kg; 50 and 100 mg/kg; 10 mg/kg; 6 mg/kg; 0.1, 1, and 10 mg/kg	Anti-inflammatory, autophagy, antioxidant	Improves memory and motor functions; decreases AChE, TNF-α, and caspase-3; increases BDNF and serotonin; inhibits amyloid formation; increases MT-III, promotes neurogenesis	0 alerts	[111,112,113,114,115,116,117,118,119,120,121,122]
Kaempferol	Flavonoid 286.23	**√**			Anti-inflammatory, autophagy, antioxidant, anti-amyloid	MMP inhibitor; BDNF modulation: antioxidant; reduces inflammatory cytokines, COX-2, HMGB1/TLR4; anti-AChE; increases dopamine; inhibits Abeta accumulation	0 alerts	[123,124,125,126,127,128,129,130,131]
Lanosterol	Phytosterol 426.71		(MPP+)-induced cell death in the PD cellular model	0.5 mM	Autophagy	Suppresses the buildup of misfolded protein aggregations/sequestosomes; promotes autophagy; mitochondrial depolarization	0 alerts	[132,133,134,135]
Leucoanthocyanins	Anthocyanins 242.26	**√**	Kainate-induced learning impairment in rats; LPS-treated adult mice; BV-2 cells	2%; 24 mg/kg; 50 and 100 μg/ml	Anti-inflammatory, autophagy, antioxidant	Modulates the PI3K/Akt/Nrf2/HO-1 pathway; COX-2/mPGES-1; promotes autophagy by upregulating AMPK–mTOR	0 alerts	[136,137,138,139,140,141,142,143,144,145,146,147]
Limonene	Terpene 136.24	**√**	Aβ-induced in vitro model of AD; scopolamine-induced amnesia rat model; subchronic effects in rats	10 μg/mL; MO: 1% and 3%; 5, 25, and 50 mg/kg	Anti-inflammatory	Improves cognition; decreases MDA, increases SOD, GSH; anti-AChE and BChE; anti-inflammatory; increases GABA	0 alerts	[148,149,150,151,152,153,154]
Lupeol	Phytosterol 426.72	**√**	Acetic acid-induced writhing, formalin test, carrageenan-induced hyperalgesia, and post-operative pain model; Aβ-induced oxidative stress in mice; TBI mouse model; cerebellar cultures	25, 50, and 100 mg/kg; 50 mg/kg; 50 mg/kg; 0.1 µM	Anti-inflammatory, antioxidant	MAPK/JNK pathway; downregulates BACE-1, upregulates proinflammatory cytokines; downregulates TNF, iNOS, NLRP3; upregulates GDNF and SHH–GLI signaling	0 alerts	[164,165,166,167,168,169,170]
*myo*-Inositol	Carbocyclic sugar 180.16	**√**	Kainic acid-induced epilepsy rat model; ischemic stroke injury in animals exposed to tobacco smoke; i streptozotocin-induced mice	0.1 μCi/mL; 30 mg/kg		Improved memory and motor functions; anticonvulsant	0 alerts	[171,172,173,174,175,176,177]
Myricetin	Flavonoid 318.23	**√**	Aβ-induced in vitro model of AD; primary neuron cultures	5 μM; 1 and 10 μM	Anti-inflammatory, autophagy, antioxidant, anti-amyloid	Decreases NF-κB and AMPK/SIRT1 signaling; reduces the levels of inflammatory mediators; autophagy; metal ion chelation; reduces A beta, anti-AChE; restores mitochondrial dysfunction	1 alert	[178,179,180,181,182,183,184,185,186,187,188,189,190]
Pinene	Terpene 136.24	**√**	Aβ-induced rat model; PC-12 cells; focal ischemic stroke model of rats; cerebral ischemia–reperfusion in rats	50 mg/kg; 10 and 25 µM; 100 mg/kg; 100 mg/kg	Anti-inflammatory, autophagy, antioxidant, anti-amyloid	Improves cognition; increases SOD, GSH, GPX, HO-1; suppresses the TNF-α/NF-κB pathway; increases the expression of choline acetyltransferase, Bcl-1, muscarinic receptors, nAChR, BDNF, and antioxidant transcription factors; decreases Bax, caspase-3	0 alerts	[191,192,193,194,195,196]
Psoralen	Coumarin 186.16		Ccopolamine-induced amnesia in rats; in vitro, in silico	0.1 and 0.3 mg/kg	Enzyme inhibition	Anti-AChE; anti-MAO	0 alerts	[197,198,199,200]
Quercetin	Flavonoid 302.23	**√**	Primary neuron cultures; MitoPark PD model; 3-NP-induced HD model	20 μM; 20 and 40 μM; 25 and 175 mg/kg; 25 mg/kg	Anti-inflammatory, antioxidant	BACE-1 inhibitor, decreases proinflammatory cytokines, increases ATP, CAT, SOD; affects PON2, Nrf2–ARE, PI3K, JNK/ERK, TNF-α, SIRT1, CREB, MAPK, NF-κB, AMPK, PGC-1α	1 alert	[201,202,203,204,205,206,207,208,209,210,211,212,213,214,215,216]
Rhein	Anthraquinone 284.22	**√**	CCI rats; I/R rats; TBI rat model; APP/PS1 mouse model of AD	12 mg/kg; 50 and 100 mg/kg; 100 mg/kg; 10 mg/kg	Anti-inflammatory, antioxidant	Increases SOD, GSH, CAT, GSH/GSSG, GSH-Px; enhances Bcl-2; decreases Bax, caspase-3, and ROS, proinflammatory cytokines; activation of the SIRT1/PGC-1α pathway; inhibits the NADPH oxidase/ROS/ERK/MMP-9 signaling pathway	1 alert	[163,217,218,219,220,221]
Rutin	Flavonoid-3-o-glycosides 610.51	**√**	Tau-P301S mouse model; 6-OHDA-induced rat model of PD; *Caenorhabditis elegans* model of HD	100 μL; 25 mg/kg; 15–120 μM	Anti-inflammatory, anti-amyloid, antioxidant	Improves memory, reduces Aβ oligomerization, oxidative stress, neurotoxicity, and neuroinflammation; reduces tau; protects dopaminergic neurons; insulin/insulin-like growth factor I pathway	1 alert	[228,229,230,231,232,233,234,235]
Stigmasterol	Phytosterol 412.69	**√**	SH-SY5Y cells; BV2 cells; scopolamine-induced memory loss in mice	1 μM; 10 and 20 μM; 3, 10, and 30 mg/kg; 10 mg/kg	Anti-inflammatory, antioxidant, anti-amyloid	Anti-AChE; reduces amyloid plaques; reduces ROS; modulates the SIRT1–FoxO3a pathway; inhibits proinflammatory cytokines; represses NF-κB and NLRP3 signaling by AMPK activation; NMDA activation; ERK/CREB activation; improves memory and LTP	0 alerts	[239,240,241,242,243,244,245,246]
Synephrine	Biogenic amine 167.21		Pre-workout; in vitro	20 mg	Enzyme inhibition	Anti-BChE and anti-AChE activity; improves the cognitive function	0 alerts	[247,248,249]
β-Caryophyllene	Sesquiterpene 204.36	**√**	APP/PS1 mice; BV2 cells; I/R injury mouse model	48 mg/kg; 5 μM; 24 and 72 mg/kg	Anti-inflammatory, autophagy, antioxidant	Anti-BACE and anti-AChE activity; increases the expression of Bcl-2, beclin-1, CB2R; decreases p62; decreases ROS and proinflammatory cytokines	0 alerts	[250,251,252,253,254]
β-Sitosterol	Phytosterol 414.71	**√**	In vitro; HT22 cells and primarily cultured hippocampal cells; LPS- induced BV2 cells	15 µM; 8 and 16 µM	Anti-inflammatory, antioxidant	Antioxidant; anti-AChE and BChE; prevents plaque deposition; modulates the PI3K/GSK-3β pathway; increases ΔΨm and ATP; decreases the expression of IL-6, iNOS, TNF-α, COX-2, IκB, NF-κB, ERK/p38	0 alerts	[255,256,257,258,259,260,261,262]

**√** indicate that the compound can cross BBB.

## Data Availability

Not applicable.

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
