# Peer review of "Mechanistic Insights into the Neuroprotective Potential of Sacred Ficus Trees"

_nutrients, 2022, doi:10.3390/nu14224731_

Round 1
Reviewer 1 Report
Shim and colleagues have attempted to review the literature on the effects of the various compounds derived from two Ficus trees in the context of some neurodegenerative diseases. Unfortunately, as presented, the review is not publishable because it has been written as a draft; it is noninformative; it is biased; it is misleading; it is misguided; it is not critical. I suggest a few points for the authors to consider before resubmitting their work for consideration as a publication. I list these points irrespective of their hierarchy of importance.
1. Nomenclature: Please note that the WHO-designated nomenclature for Aβ is ‘amyloid β-protein’. Please use this throughout the manuscript. ‘Aβ’ as an abbreviation is fine.
2. Logically wrong conventions: Despite being the commonly used and commonly accepted ‘misnomer’, the eponymous term ‘Alzheimer’s Disease’ (AD) is logically incorrect. Historically, the disease was discovered by Alois Alzheimer; the disease was not ‘his own’ personal disease. Because of the eponymous convention, using the possessive form (apostrophe plus ‘s’ or genitive ‘s’) is wrong but has become perpetual in the English Scientific literature by our great peers. Many though have avoided it. The Australian Manual of Scientific Style and The Chicago Manual of Style advise against the use of the possessive form. I suggest taking their editorial advice and applying it throughout the text. The same applies to Parkinson Disease and Huntington Disease.
3. The authors must provide sufficient details when referring to model studies using the various extracts, or the compounds used before extrapolating such findings to applications in human disease. For example, when referring to studies which have used in vitro cell models or in vivo animal models, provide sufficient detail about the studies, their aims, their methods, their findings, and the limitations of the models used, instead of merely listing and summarising them. Such information will be beneficial to the reader and the corresponding summaries should be tabulated. The authors must discuss the limitations or advantages of such studies, or the models used and provide information one by one, especially comparing the concentrations of the compounds used, the purity of the compounds, or the types of assays used in the referred studies. For example, information on the bioavailability of the compounds is important.
4. The authors should discuss the implications of using a type of extract that contains mixtures of various compounds, either characterised or not. For example, what does using a mixture mean in terms of specificity of demonstrating a cause-and-effect relationship? (See below comment)
5. My major concern in the approach of this review is the lack of criticism on the use of herbal extracts of various types from various parts of the plants. Using a mixture does not allow attributing the activities observed to a single chemical. I understand that the chemicals in these mixtures may or may not have all been characterised in terms of identity or proportional concentration (such details should be clearly discussed in this review). This fact complicates the situation because some plant-derived chemicals and polyphenols have been reported to be functioning as pan-assay-interference compounds (PAINS) and as invalid metabolic panacea (IMP) (1, 2). For example, Nelson et al. have raised some considerable concerns about the experimental use of curcumin and its medicinal chemistry. This discussion is worth reading before attempting any experiment using curcumin or other herbal mixtures. The concerns about the PAINS and IMPS characteristics of curcumin include its impurity, chemical instability, its being a poor lead compound, its off-target properties, and its poor pharmacokinetic and pharmacodynamic properties (1, 2). Nelson et al. have also critically evaluated the clinical studies and trials using curcumin (1, 2). According to Neslon et al. the distribution of curcumin to various tissues and organs is very limited (1, 2). I would like the authors to review the body of the work discussed by Nelson et al. and discuss their work in the context of the compounds and chemicals that they have listed in the review. Quercetin, for example, has been claimed to function as a PAINS. A major question to consider and discuss would be, ‘Do any of the components listed in the review have the potential to function as PAINS or IMPs?’ In the same context, please see a commentary by Baell et al. (3) and another by Baker (4). Baker mentions that between 2009 to 2017, at least 15 articles on curcumin were retracted. At once, the retraction-watch database returned 47 retracted articles with the word ‘curcumin’ in their title—with various reasons for retractions. Do similar concerns apply to using herbal mixtures? Please discuss this in detail in the manuscript.
6. Related to the comment above, I suggested the authors must consult the PAINS database to find out information on the compounds listed and discussed. If any information is available that the discussed compounds function as PAINS, then mention the information in the text and discuss. This is the link to the database: The PAINS Remover (https://www.cbligand.org/PAINS/). This is important for future generations of medicinal chemists so they don’t waste time and resources on compounds that may be problematic or may not have any therapeutic potential in treating human disease. In fact, the list of effects that the reviewed compounds can have suggests to me that they are acting as PAINS or IMPs.
7. As an example of being critical, unbiased, and informative, the authors must refer to the literature highlighting the controversies around the ‘amyloid-cascade hypothesis’. The ‘amyloid-cascade hypothesis’ and the ‘oligomer-cascade hypothesis’ underlying Alzheimer disease have been debated (5-10). This is important to be contextualised when referring to previous literature discussing the inhibitory roles of the plant compounds on the aggregation of Aβ. Besides, many treatments designed based on the tenets of the ‘amyloid-cascade hypotheses’ have failed and are likely to continue failing (11).
8. In the literature that I cited and exemplified here or by the same authors, limitations of the animal models have been discussed in the context of neurodegenerative diseases (especially Alzheimer Disease). The authors must consider these debates while formulating their views in this review and mainly present an informative debate on the existing literature instead of merely listing the papers or the literature that show an effect of a plant compound.
9. Referring to historical myths regarding the plants in the context of religion or the bible is not solid scientific justification; for example, see the reference to Adam and Eve that they covered their bodies with the leaves of the Ficus trees. Present scientific evidence that this past story is correct. Did Adam come about first or the Ficus tree?
10. The text suffers from many English mistakes including punctuation, number disagreement, and unclear and vague statements (e.g., the statement in lines 59–60). I refrain from counting and listing the English errors here in this peer-review but give one example: statements of fact must be written in the present tense. Therefore, the authors must seek help from an academic English editor before resubmitting the text.
References:
1. Nelson KM, Dahlin JL, Bisson J, Graham J, Pauli GF, Walters MA. The Essential Medicinal Chemistry of Curcumin. J Med Chem. 2017;60(5):1620-1637. https://doi.org/10.1021/acs.jmedchem.6b00975.
2. Nelson KM, Dahlin JL, Bisson J, Graham J, Pauli GF, Walters MA. Curcumin May (Not) Defy Science. ACS Med Chem Lett. 2017;8(5):467-470. https://doi.org/10.1021/acsmedchemlett.7b00139.
3. Baell J, Walters MA. Chemistry: Chemical con artists foil drug discovery. Nature. 2014;513(7519):481-483. https://doi.org/10.1038/513481a.
4. Baker M. Deceptive curcumin offers cautionary tale for chemists. Nature. 2017;541(7636):144-145. https://doi.org/10.1038/541144a.
5. Clark IA, Vissel B. Amyloid b: one of three danger-associated molecules that are secondary inducers of the proinflammatory cytokines that mediate Alzheimer's disease. Br J Pharmacol. 2015;172(15):3714-3727. https://doi.org/10.1111/bph.13181.
6. Clark IA, Vissel B. Excess cerebral TNF causing glutamate excitotoxicity rationalizes treatment of neurodegenerative diseases and neurogenic pain by anti-TNF agents. J Neuroinflammation. 2016;13(1):236. https://doi.org/10.1186/s12974-016-0708-2.
7. Clark IA, Vissel B. Therapeutic implications of how TNF links apolipoprotein E, phosphorylated tau, a-synuclein, amyloid-b and insulin resistance in neurodegenerative diseases. Br J Pharmacol. 2018;175(20):3859-3875. https://doi.org/10.1111/bph.14471.
8. Clark IA, Vissel B. Broader insights into understanding tumor necrosis factor and neurodegenerative disease pathogenesis infer new therapeutic approaches. J Alzheimers Dis. 2021;79(3):931-948. https://doi.org/10.3233/JAD-201186.
9. Morris GP, Clark IA, Vissel B. Inconsistencies and controversies surrounding the amyloid hypothesis of Alzheimer's disease. Acta Neuropathol Commun. 2014;2:135. https://doi.org/10.1186/s40478-014-0135-5.
10. Morris GP, Clark IA, Vissel B. Questions concerning the role of amyloid-b in the definition, aetiology and diagnosis of Alzheimer's disease. Acta Neuropathol. 2018;136(5):663-689. https://doi.org/10.1007/s00401-018-1918-8.
11. Lowe D. In the Pipeline: Alzheimer’s disease. 2021. Available from: https://blogs.sciencemag.org/pipeline/archives/category/alzheimers-disease. [Accessed 21 February 2021].
Author Response
Reviewer 1
Comments and Suggestions for Authors
Shim and colleagues have attempted to review the literature on the effects of the various compounds derived from two Ficus trees in the context of some neurodegenerative diseases. Unfortunately, as presented, the review is not publishable because it has been written as a draft; it is noninformative; it is biased; it is misleading; it is misguided; it is not critical. I suggest a few points for the authors to consider before resubmitting their work for consideration as a publication. I list these points irrespective of their hierarchy of importance.
We wish to express our thanks to the reviewer for the input and suggestions on our manuscript. We have incorporated the changes in the manuscript in red fonts.
1.Nomenclature: Please note that the WHO-designated nomenclature for Aβ is ‘amyloid β-protein’. Please use this throughout the manuscript. ‘Aβ’ as an abbreviation is fine.
√ We have made corrections as suggested by the reviewer.
2. Logically wrong conventions: Despite being the commonly used and commonly accepted ‘misnomer’, the eponymous term ‘Alzheimer’s Disease’ (AD) is logically incorrect. Historically, the disease was discovered by Alois Alzheimer; the disease was not ‘his own’ personal disease. Because of the eponymous convention, using the possessive form (apostrophe plus ‘s’ or genitive ‘s’) is wrong but has become perpetual in the English Scientific literature by our great peers. Many though have avoided it. The Australian Manual of Scientific Style and The Chicago Manual of Style advise against the use of the possessive form. I suggest taking their editorial advice and applying it throughout the text. The same applies to Parkinson Disease and Huntington Disease.
√ We have used scientific terms according to the nomenclature used in the literature. We are ready to change the term if the Editors think the same way.
3.The authors must provide sufficient details when referring to model studies using the various extracts, or the compounds used before extrapolating such findings to applications in human disease. For example, when referring to studies which have used in vitro cell models or in vivo animal models, provide sufficient detail about the studies, their aims, their methods, their findings, and the limitations of the models used, instead of merely listing and summarising them. Such information will be beneficial to the reader and the corresponding summaries should be tabulated. The authors must discuss the limitations or advantages of such studies, or the models used and provide information one by one, especially comparing the concentrations of the compounds used, the purity of the compounds, or the types of assays used in the referred studies. For example, information on the bioavailability of the compounds is important.
√ We have thoroughly revised the manuscript and included more details for the experimental work in the tables and text.
4.The authors should discuss the implications of using a type of extract that contains mixtures of various compounds, either characterised or not. For example, what does using a mixture mean in terms of specificity of demonstrating a cause-and-effect relationship? (See below comment).
√ The authors are thankful to the reviewer for the critical evaluation of the manuscript. In the manuscript, we have divided the tables as “Table 1 for the activity of plant extracts” where the ativity was monitored using the whole extract, and “Table 2 for neuroprotection by the phytocompounds.” We have now revised the table according to the comments (5 & 6) by the reviewer
5. My major concern in the approach of this review is the lack of criticism on the use of herbal extracts of various types from various parts of the plants. Using a mixture does not allow attributing the activities observed to a single chemical. I understand that the chemicals in these mixtures may or may not have all been characterised in terms of identity or proportional concentration (such details should be clearly discussed in this review). This fact complicates the situation because some plant-derived chemicals and polyphenols have been reported to be functioning as pan-assay-interference compounds (PAINS) and as invalid metabolic panacea (IMP) (1, 2). For example, Nelson et al. have raised some considerable concerns about the experimental use of curcumin and its medicinal chemistry. This discussion is worth reading before attempting any experiment using curcumin or other herbal mixtures. The concerns about the PAINS and IMPS characteristics of curcumin include its impurity, chemical instability, its being a poor lead compound, its off-target properties, and its poor pharmacokinetic and pharmacodynamic properties (1, 2). Nelson et al. have also critically evaluated the clinical studies and trials using curcumin (1, 2). According to Neslon et al. the distribution of curcumin to various tissues and organs is very limited (1, 2). I would like the authors to review the body of the work discussed by Nelson et al. and discuss their work in the context of the compounds and chemicals that they have listed in the review. Quercetin, for example, has been claimed to function as a PAINS. A major question to consider and discuss would be, ‘Do any of the components listed in the review have the potential to function as PAINS or IMPs?’ In the same context, please see a commentary by Baell et al. (3) and another by Baker (4). Baker mentions that between 2009 to 2017, at least 15 articles on curcumin were retracted. At once, the retraction-watch database returned 47 retracted articles with the word ‘curcumin’ in their title—with various reasons for retractions. Do similar concerns apply to using herbal mixtures? Please discuss this in detail in the manuscript.
√ Thanks for raising an important point. Pan-assay-interference compounds (PAINS) have false activities in vitro and in vivo, which can waste time, effort, and resources in research. Phyto-compounds extracted from Ficus trees may also have the potential of PAINS and we agreed with the importance of communicating this to the readers. The contents of this were stated in the manuscript and table2.
6. Related to the comment above, I suggested the authors must consult the PAINS database to find out information on the compounds listed and discussed. If any information is available that the discussed compounds function as PAINS, then mention the information in the text and discuss. This is the link to the database: The PAINS Remover (https://www.cbligand.org/PAINS/). This is important for future generations of medicinal chemists so they don’t waste time and resources on compounds that may be problematic or may not have any therapeutic potential in treating human disease. In fact, the list of effects that the reviewed compounds can have suggests to me that they are acting as PAINS or IMPs.
√ Thanks for your suggestion. We applied the SwissADME server for Medicinal chemistry analysis of PAINS. The predicted structural alerts of each compound were indicated in Table 2.
7. As an example of being critical, unbiased, and informative, the authors must refer to the literature highlighting the controversies around the ‘amyloid-cascade hypothesis’. The ‘amyloid-cascade hypothesis’ and the ‘oligomer-cascade hypothesis’ underlying Alzheimer disease have been debated (5-10). This is important to be contextualised when referring to previous literature discussing the inhibitory roles of the plant compounds on the aggregation of Aβ. Besides, many treatments designed based on the tenets of the ‘amyloid-cascade hypotheses’ have failed and are likely to continue failing (11).
√ Thank you very much for your suggestions. We have incorportated it in the text.
8. In the literature that I cited and exemplified here or by the same authors, limitations of the animal models have been discussed in the context of neurodegenerative diseases (especially Alzheimer Disease). The authors must consider these debates while formulating their views in this review and mainly present an informative debate on the existing literature instead of merely listing the papers or the literature that show an effect of a plant compound.
√ We have incorporated the reviewer’s suggestion in the “Conclusion”.
9. Referring to historical myths regarding the plants in the context of religion or the bible is not solid scientific justification; for example, see the reference to Adam and Eve that they covered their bodies with the leaves of the Ficus trees. Present scientific evidence that this past story is correct. Did Adam come about first or the Ficus tree?
√ We are not justifying these historical/mythological facts scientifically as the review is directed toward describing the neuroprotective mechanism of these trees. The purpose of citing historical and spiritual references in the Introduction is to denote the significance and perpetuity of the Ficus trees. To reply to the interesting question regarding Adam we would like to quote the Biblical text Genesis chapter 3:7 “Then the eyes of both of them were opened, and they realized they were naked; so they sewed fig leaves together and made coverings for themselves.”
10. The text suffers from many English mistakes including punctuation, number disagreement, and unclear and vague statements (e.g., the statement in lines 59–60). I refrain from counting and listing the English errors here in this peer-review but give one example: statements of fact must be written in the present tense. Therefore, the authors must seek help from an academic English editor before resubmitting the text.
√ The manuscript has been revised thoroughly for content and grammatical errors as suggested by the reviewer.
References:
1. Nelson KM, Dahlin JL, Bisson J, Graham J, Pauli GF, Walters MA. The Essential Medicinal Chemistry of Curcumin. J Med Chem. 2017;60(5):1620-1637. https://doi.org/10.1021/acs.jmedchem.6b00975.
2. Nelson KM, Dahlin JL, Bisson J, Graham J, Pauli GF, Walters MA. Curcumin May (Not) Defy Science. ACS Med Chem Lett. 2017;8(5):467-470. https://doi.org/10.1021/acsmedchemlett.7b00139.
3. Baell J, Walters MA. Chemistry: Chemical con artists foil drug discovery. Nature. 2014;513(7519):481-483. https://doi.org/10.1038/513481a.
4. Baker M. Deceptive curcumin offers cautionary tale for chemists. Nature. 2017;541(7636):144-145. https://doi.org/10.1038/541144a.
5. Clark IA, Vissel B. Amyloid b: one of three danger-associated molecules that are secondary inducers of the proinflammatory cytokines that mediate Alzheimer's disease. Br J Pharmacol. 2015;172(15):3714-3727. https://doi.org/10.1111/bph.13181.
6. Clark IA, Vissel B. Excess cerebral TNF causing glutamate excitotoxicity rationalizes treatment of neurodegenerative diseases and neurogenic pain by anti-TNF agents. J Neuroinflammation. 2016;13(1):236. https://doi.org/10.1186/s12974-016-0708-2.
7. Clark IA, Vissel B. Therapeutic implications of how TNF links apolipoprotein E, phosphorylated tau, a-synuclein, amyloid-b and insulin resistance in neurodegenerative diseases. Br J Pharmacol. 2018;175(20):3859-3875. https://doi.org/10.1111/bph.14471.
8. Clark IA, Vissel B. Broader insights into understanding tumor necrosis factor and neurodegenerative disease pathogenesis infer new therapeutic approaches. J Alzheimers Dis. 2021;79(3):931-948. https://doi.org/10.3233/JAD-201186.
9. Morris GP, Clark IA, Vissel B. Inconsistencies and controversies surrounding the amyloid hypothesis of Alzheimer's disease. Acta Neuropathol Commun. 2014;2:135. https://doi.org/10.1186/s40478-014-0135-5.
10. Morris GP, Clark IA, Vissel B. Questions concerning the role of amyloid-b in the definition, aetiology and diagnosis of Alzheimer's disease. Acta Neuropathol. 2018;136(5):663-689. https://doi.org/10.1007/s00401-018-1918-8.
11. Lowe D. In the Pipeline: Alzheimer’s disease. 2021. Available from: https://blogs.sciencemag.org/pipeline/archives/category/alzheimers-disease. [Accessed 21 February 2021].
Reviewer 2 Report
The review by Kyu Hwan Shim et al. focuses on the neuroprotective potential of two specific plants: F.Religiosa and F.Bengalensis. This work overviews the researches on the activity exerted by discrete plant parts (leaves, roots, etc). Then, the Authors provide details on the effects induced by the main phytochemicals present.
1) The original point of this work is the description of the neuroprotective action of the phytocomplexes, that would definitely deserve a greater extent of dissertation. Especially, if the Authors aim at identifying novel potential points that could play a role in experimental models of neurodegenerative disorders.
2) What was the detailed methodology used by the Authors? A novel paragraph should be included expanding the information provided at the end of paragraph 1, in order to describe the methods of literature search, stating which databases were used, what keywords, how many publications were included over the total amount found.
3) An in-depth revision by a Native Speaker is strongly suggested
Author Response
Reviewer 2
Comments and Suggestions for Authors
The review by Kyu Hwan Shim et al. focuses on the neuroprotective potential of two specific plants: F.Religiosa and F.Bengalensis. This work overviews the researches on the activity exerted by discrete plant parts (leaves, roots, etc). Then, the Authors provide details on the effects induced by the main phytochemicals present.
Our sincere thanks to the reviewer for the comments and suggestions on the improvement of our manuscript. All changes in the manuscript are indicated in red fonts.
1) The original point of this work is the description of the neuroprotective action of the phytocomplexes, that would definitely deserve a greater extent of dissertation. Especially, if the Authors aim at identifying novel potential points that could play a role in experimental models of neurodegenerative disorders.
√ We have thoroughly revised the manuscript and included more details for the experimental work in the tables and text.
2) What was the detailed methodology used by the Authors? A novel paragraph should be included expanding the information provided at the end of paragraph 1, in order to describe the methods of literature search, stating which databases were used, what keywords, how many publications were included over the total amount found.
√ As suggested by the reviewer, the changes have been made in the manuscript in red font.
3) An in-depth revision by a Native Speaker is strongly suggested
√ The manuscript has been revised thoroughly for content and grammatical errors as suggested by the reviewer.
Round 2
Reviewer 1 Report
It seems some of my suggestions were not thoroughly considered or implemented by the authors. I can interpret this to have been caused by the language barrier or misunderstanding of my concerns. Hope that these concerns will be taken into account during the production stage of the manuscript. For example, see the specific notation of 'amyloid β-protein' instead of 'amyloid beta protein' that is used in the text. Attention to detail is critical. The same applies to my suggestion about using the accepted scientific style of disease nomenclature. The authors should take initiative instead of relying on editors. Good luck with your future work.
Reviewer 2 Report
authors have satisfactory addressed all the issues